# Plethora of Antibiotics Usage and Evaluation of Carbapenem Prescribing Pattern in Intensive Care Units: A Single-Center Experience of Malaysian Academic Hospital

**DOI:** 10.3390/antibiotics11091172

**Published:** 2022-08-30

**Authors:** Chee Lan Lau, Petrick Periyasamy, Muhd Nordin Saud, Sarah Anne Robert, Lay Yen Gan, Suet Yin Chin, Kiew Bing Pau, Shue Hong Kong, Farah Waheeda Tajurudin, Mei Kuen Yin, Sheah Lin Ghan, Nur Jannah Azman, Xin Yun Chua, Poy Kei Lye, Stephanie Wai Yee Tan, Dexter Van Dort, Ramliza Ramli, Toh Leong Tan, Aliza Mohamad Yusof, Saw Kian Cheah, Wan Rahiza Wan Mat, Isa Naina-Mohamed

**Affiliations:** 1Pharmacoepidemiology and Drug Safety Unit, Department of Pharmacology, Faculty of Medicine, Universiti Kebangsaan Malaysia, Cheras, Kuala Lumpur 56000, Malaysia; 2Pharmacy Department, Hospital Canselor Tuanku Muhriz, Cheras, Kuala Lumpur 56000, Malaysia; 3Medical Department, Faculty of Medicine, Universiti Kebangsaan Malaysia, Cheras, Kuala Lumpur 56000, Malaysia; 4Department of Medical Microbiology and Immunology, Faculty of Medicine, Universiti Kebangsaan Malaysia, Cheras, Kuala Lumpur 56000, Malaysia; 5Department of Emergency Medicine, Faculty of Medicine, Universiti Kebangsaan Malaysia, Cheras, Kuala Lumpur 56000, Malaysia; 6Department of Anesthesiology & Intensive Care, Faculty of Medicine, Universiti Kebangsaan Malaysia, Cheras, Kuala Lumpur 56000, Malaysia

**Keywords:** carbapenems, defined daily dose, antibiotics utilization, intensive care, empiric

## Abstract

Excessive antibiotic consumption is still common among critically ill patients admitted to intensive care units (ICU), especially during the coronavirus disease 2019 (COVID-19) period. Moreover, information regarding antimicrobial consumption among ICUs in South-East Asia remains scarce and limited. This study aims to determine antibiotics utilization in ICUs by measuring antibiotics consumption over the past six years (2016–2021) and specifically evaluating carbapenems prescribed in a COVID-19 ICU and a general intensive care unit (GICU) during the second year of the COVID-19 pandemic. (2) Methods: This is a retrospective cross-sectional observational analysis of antibiotics consumption and carbapenems prescriptions. Antibiotic utilization data were estimated using the WHO Defined Daily Doses (DDD). Carbapenems prescription information was extracted from the audits conducted by ward pharmacists. Patients who were prescribed carbapenems during their admission to COVID-19 ICU and GICU were included. Patients who passed away before being reviewed by the pharmacists were excluded. (3) Results: In general, antibiotics consumption increased markedly in the year 2021 when compared to previous years. Majority of carbapenems were prescribed empirically (86.8%). Comparing COVID-19 ICU and GICU, the reasons for empirical carbapenems therapy in COVID-19 ICU was predominantly for therapy escalation (64.7% COVID-19 ICU vs. 34% GICU, *p* < 0.001), whereas empirical prescription in GICU was for coverage of extended-spectrum beta-lactamases (ESBL) gram-negative bacteria (GNB) (45.3% GICU vs. 22.4% COVID-19 ICU, *p* = 0.005). Despite microbiological evidence, the empirical carbapenems were continued for a median (interquartile range (IQR)) of seven (5–8) days. This implies the need for a rapid diagnostic assay on direct specimens, together with comprehensive antimicrobial stewardship (AMS) discourse with intensivists to address this issue.

## 1. Introduction

Antibiotics have been prescribed in 70% of ICU patients due to the high prevalence of suspected or proven infection [1]. Since the outbreak of COVID-19, the hospitalization rate has increased along with an increased tendency of antibiotics prescription. A retrospective study in Malaysia during the early phase of the pandemic found a lower antibiotic usage at a prevalence of only 17.1%, in contrast to findings by two systematic reviews [2,3], though it was observed that ICU/HDU admissions were 2.73 times more likely to be prescribed antibiotics [4]. However, no details on antibiotic dosage and duration were analyzed.

A systematic review of 38 studies consisting of 2715 ICU admissions found a similar frequency of antibiotics prescription at 71%. Yet, incidences of bacterial infections were reported in only 30.8% of the studies reviewed. Furthermore, 69.2% of the antibiotics prescribed were empirical without strong evidence of bacterial infection [5]. In a review by Pasero et al. [6], hospital-acquired infection among COVID-19 patients developed 10–15 days after ICU admissions. However, extensive empirical antibiotics were prescribed, along with prolonged ICU stay leading to the surge of multidrug resistance (MDR) microorganisms, with incidence ranging from 32% to 50%. These data only reflected the use of antibiotics during the first year of the pandemic, and studies on the prescription pattern among critically ill patients in developing countries and the South-East Asia region are scarce. In addition, little is known about the duration of exposure to the prescribed antibiotic(s), which is crucial for antibiotic resistance development [7].

Antimicrobial resistance (AMR) has been a global health threat declared by World Health Organization (WHO) since 2015 [8]. With the high prevalence of antibiotic prescription and infection rates, ICU may potentially be the driver of resistance in hospitals [9]. Furthermore, an increase in antimicrobials resistance (AMR) in ICUs was observed since the COVID-19 pandemic, owing to the compromise in infection control and excessive antimicrobials use [10]. Carbapenems consumption has a positive correlation with increased resistance to carbapenems among gram-negative organisms such as *Acinetobacter baumannii*, *Pseudomonas aeruginosa*, and *Enterobacterales* [11,12,13]. Till the year 2020, surveillance in local hospitals of Malaysia reported that resistance to meropenem was lower than 10% for most gram-negative organisms, except *Acinetobacter baumannii* (58.5%) [14]. However, it is just a matter of time before carbapenems resistance rate increases beyond 20% as seen with resistance to third-generation cephalosporins in *Klebsiella pneumoniae* [14]. Hence, local antibiotics consumption should be monitored and the reasons for empirical usage of broad-spectrum antibiotics like carbapenems should be explored. This present study attempts to determine antibiotic utilization in ICUs over the past six years and analyze the prescription of carbapenems in COVID-19 ICU and GICU during the second year of the COVID-19 pandemic.

## 2. Results

The usage of antibiotics was stable from 2016 through 2019. Comparing the year 2019 and year 2021, the total consumption of selected antibiotics (Figure 1) in both ICUs had increased from 823.9 DDD per 1000 patient days to 1307.6 DDD per 1000 patient days (Appendix A). In contrast to the increase in ceftriaxone from 117.4 to 146.9 DDD per 1000 days, amoxicillin/clavulanic acid was raised more than two-fold from 47.9 to 112.7 DDD per 1000 patient days, while ampicillin/sulbactam was raised from 140.5 to 240.3 DDD per 1000 patient days. Notably, the utilization of colistin surged and was almost 10 times higher; it increased from 2.95 to 32.04 DDD per 1000 patient days while that of polymyxin B dropped 15% from 52.6 to 44.5 DDD per patient days. Piperacillin/tazobactam consumption increased from 187.4 to 246.7 DDD per 1000 patient days, but cefepime usage increased and was more than three times higher; it went from from 46 to 134.8 DDD per 1000 patient days. Meanwhile, vancomycin utilization was also raised by 81.7%, from 52 to 94.5 DDD per 1000 patient days.

### 2.1. Carbapenems Consumption

Considering the past six years, the total admissions had dropped since 2020 and were the lowest in 2021. However, the average length of stay per patient and total patient days in both ICUs were the longest in 2021 at 8.02 days and 6229 days, respectively (Table 1). The average consumption of type-2 carbapenems in 2016 to 2019 was maintained at a median (IQR) of 153.3 (140.6–161.0) DDD per 1000 patient days. Subsequently, the usage increased by 53.6% in 2021 compared to 2019.

### 2.2. Carbapenems Prescribing in COVID-19 ICU & GICU

#### 2.2.1. Carbapenems Prescriptions

In 2021, a total of 605 carbapenems prescription requests were retrieved from the preauthorization forms, of which 159 prescriptions for 149 patients in the GICU and the COVID-19 ICU were eligible to be included (Figure 2). Meanwhile, a total of five prescriptions were excluded because they were missed, or patients passed away before being reviewed by pharmacists.

#### 2.2.2. Patients’ Demographics & Infection Control Surveillance

In 2021, there were 336 admissions to COVID-19 ICU and 231 admissions to GICU. The all-cause in-ICU mortality was higher (127, 37.8% vs. 40, 17.3%, *p* < 0.0001) and the median (IQR) length of ICU stay was longer (9 (5–15) days vs. 5 (3–10) days, *p* < 0.0001) in the COVID-19 ICU compared to the GICU.

Among patients who were prescribed carbapenems, the majority were male patients (94/149, 63.1%) with a median (IQR) age of 61 (44–69) years old. The male proportion (56/91 vs. 40/58, *p* = 0.297) and patients’ age (median (IQR): 61 (46–68) years old vs. 60 (37–71) years old, *p* = 0.806) were comparable between COVID-19 ICU and GICU. Notably, GICU had significantly more patients colonized with resistant organisms who were prescribed carbapenems (*p* = 0.003) (Table 2).

#### 2.2.3. Characteristics of Carbapenems Prescriptions

At the time of prescription, most of the carbapenems were intended for nosocomial infection (type-3) (79.9%), followed by healthcare-associated infection (type-2), and six prescriptions were for community-acquired infection (type-1). Most prescriptions were for nosocomial infections in COVID-19 ICU (83/96 vs. 44/63, *p* = 0.011). In the GICU, the majority were for healthcare-associated infections in the GICU (11/96 vs. 15/63, *p* = 0.039) (see Table 3). Meropenem accounted for most of the carbapenems prescribed across both ICUs. Overall, only 21 (13.2%) of carbapenems prescriptions were for definitive therapy according to the microbiological reports, and 86.8% were for empirical therapy.

#### 2.2.4. Empirical Carbapenems Therapy

More than half of the carbapenems were prescribed for escalation therapy, followed by the consideration of ESBL GNB risk (Table 4). Conversely, only 10 (7.2%) prescriptions were initiated after infectious disease (ID) consultation. Type-2 patients were more often prescribed for consideration of ESBL GNB risk (15/24, 62.5%, *p* < 0.001). Empirical escalation to carbapenems was often prescribed for type-3 patients (65/108, *p* = 0.001), and predominantly observed in COVID-19 ICU (55/85 vs. 18/53, *p* < 0.001). The initiation of empirical therapy considering ESBL GNB risk was more frequent (19/85 vs. 24/53, *p* = 0.005) in GICU. No association was found between reasons for empirical therapy with sites of infection. However, empirical therapy was more often intended for respiratory infections in the COVID-19 ICU (*p* = 0.017).

#### 2.2.5. Microbiological Growth & Organisms

Overall, out of 159 prescriptions, 101 (63.5%) had positive growth from cultures and 66 (41.5%) from blood cultures. The remaining 58 (36.5%) prescriptions had no growth, mixed growth, or candida species from respiratory samples or urine samples. 

Among definitive therapies, *Klebsiella pneumoniae* and *Klebsiella* spp. (14/23, 60.9%) were frequently isolated, and the majority were ESBL producers. Two *Klebsiella* isolates were carbapenemases producers (Table 5). This was followed by ESBL-producing *Escherichia coli* (5/23, 21.7%). All isolates were from type-2 and type-3 patients. Among 50 patients with rectal colonization, only eight patients (16.0%) had ESBL GNB bacteremia, compared to eight (8.9%) among 90 patients without colonization (*p* = 0.268). For the empirical prescriptions, the isolated organisms are listed in Table 5. MDR *Acinetobacter* spp. were frequently isolated, especially from COVID-19 ICU (*p* = 0.143), whereas *Enterobacterales* and *Pseudomonas aeruginosa* were isolated more often from GICU. 

#### 2.2.6. Duration of Carbapenems Therapy

During the carbapenems therapy, 55 patients passed away before completing the treatment. Overall, the median (IQR) duration of carbapenems prescriptions was seven (5–8) days. The duration of definitive therapy was significantly longer than that of empirical therapy by one day (*p* = 0.015). Compared to GICU, a shorter duration of definitive therapy (*p* = 0.463), but a longer duration of empirical therapy (*p* = 0.654) was observed in COVID-19 ICU (Table 6). In addition, among empirical prescriptions, only seven (13.0%) in COVID-19 ICU and seven (20.0%) in GICU were discontinued within three days (*p* = 0.624).

## 3. Discussion

The high prevalence of antibiotics prescription for critically ill patients admitted to ICU is common [1,15]. Interestingly, the same proportion at 70% was found during the pandemic period [5]. However, the issue of increased antibiotic consumption during the pandemic period is mostly reflected in the proportion of patients being prescribed antibiotics [2,5], where few reported the magnitude of antibiotic utilization in ICU with the measure of DDD [3], which is also an important indicator for usage trend monitoring, the impact of intervention, global comparison [16], as well as correlation with resistance trends [12,17,18]. The decrease in total ICU admissions was attributed to the opening of the COVID-19 ICU with redistribution of manpower and the closure of the operating theatre elective list during the COVID-19 pandemic. However, this was followed by longer ICU stays and higher antibiotics consumption. Notably, during the pre-pandemic period, 2016–2019, the utilization of carbapenems, vancomycin, and polymyxins consumption was found to be lower than that in surgical ICUs in Serbia (135–340, 83–64, 73–66 DDD per 1000 patient days) [17] and medical-surgical ICUs in Saudi Arabia (345.9, 180.0 and 157.1 DDD per 1000 patient days) [19], in both of which AMS interventions were absent. On the other hand, despite the different measures and denominators used, which made it difficult to perform a direct comparison, this study found an increase in annual consumption of most antibiotics during the pandemic year, similar to a Brazilian ICUs [20], yet contrary to the findings in Spanish ICUs that observed a decrease in meropenem and piperacillin/tazobactam [21]. The relatively lower prevalence of antibiotic prescription found by the study in Malaysia [4] could have been masked by various populations across disciplines. Nevertheless, the trajectory increases in antibiotic usage demanded the need to probe into the prescription rationale and the difference between the COVID-19 ICU and the GICU.

Following the CLSI revision in 2020 on the breakpoint and questioning the clinical value of polymyxin(s) [22,23], empirical use of polymyxins was discouraged and the restriction on polymyxins was further enhanced to definitive therapy only with microbiological evidence, in addition to consent by an ID consultant. Therefore, the consumption in 2021 likely reflected the definite use of polymyxins according to culture reports. In this hospital, intravenous polymyxin B was the primary polymyxin of choice for infection due to carbapenem-resistant gram-negative organisms (CRGNB) [24], whereas colistin was preferred and used intravenously for urinary tract infection [24] or as an inhalation therapy for pneumonia [25]. The sharp increase in colistin implied the higher tendency to treat carbapenem-resistant organisms such as *Acinetobacter baumannii* (CRAb) isolated from respiratory cultures, although the clinical significance is debatable, especially in COVID-19 infected critically ill patients [26]. In contrast, the use of polymyxin B did not increase but was similar to the previous years; it was persistent for carbapenem-resistant gram-negative organisms isolated from blood cultures.

Carbapenems are broad-spectrum antibiotics belonging to the WATCH group under the WHO AWaRe classification, which should be the focus of stewardship [27]. Furthermore, the resistance rate among gram-negative organisms towards carbapenems is on the rise globally, which is attributed to carbapenems use [28,29]. In the setting of limited human resources in our hospital, efforts were, therefore, mainly focused on carbapenems instead of targeting all antibiotics. Following the introduction of local ICU antibiotic treatment protocol (Appendix A) and a weekly visit of ID consultants to ICUs since 2016, the consumption of carbapenems in the ICU was maintained at lower than 200 DDD per 1000 patient days. However, the weekly ID rounds were halted in 2020 due to the pandemic and antibiotics usage has increased since then.

To our knowledge, this was the first study to compare the carbapenems prescription pattern between the COVID-19 ICU and the GICU. Meropenem was the preferred agent used, as it had better activities on gram-negative bacteria and central nervous system penetration [30]. This is consistent with the observations in the recent systematic reviews [3,5]. Good compliance to local treatment protocol (Appendix A) was observed as carbapenems were prescribed mainly for type-3 patients who were at risk of infection by multi-drug resistant organisms [25]. Broad spectrum antibiotics were recommended by the last surviving sepsis guideline for the critically ill, as failure to cover possible pathogens in sepsis will lead to higher mortality [31,32]. Patient types were determined at the point of carbapenems prescription; hence, a higher proportion of type-3 patients in the COVID-19 ICU was likely a result of longer ICU stay. Predictably, carbapenems were prescribed empirically in most cases. The fraction of empirical carbapenems prescriptions from the GICU alone was still higher than the reported 66.1% in French ICUs [33], though the latter was studied during the pre-COVID-19 pandemic period. A similar proportion in either ICU indicated that clinicians were practicing high rates of empirical carbapenems, despite the negative culture or growth of the organism(s) susceptible to narrower spectrum beta-lactam antibiotics or alternatives. Although carbapenems were the recommended empirical choice for ICU patients with severe sepsis [13], only a small percentage of prescriptions had positive growth of ESBL-producing *Enterobacterales*, which were predominantly *Klebsiella* spp., similar to another tertiary hospital in the same region in Malaysia [18].

The reasons for empirical prescription differed between the COVID-19 ICU and the GICU, associated with the distribution of the patient types. Rectal colonization with ESBL/MDR GNB was listed as a risk factor for infection [34,35,36]. Therefore, this drove the carbapenems prescription [36], as seen with the GICU. However, the clinical value is debatable as the positive predictive value is up to 50%, and the screening is unreliable for ICU patients [37]. The current risk stratification for predicting ESBL GNB infection was derived from criteria commonly listed in other predicting models with the same flaw of lacking external validity [38]. To have a better balance between the consequence of carbapenems exposure and management of infection, a validated scoring system is urgently needed to allow more objective judgment.

Empirical carbapenems prescription was seen to be mainly driven by the intention to escalate therapy in the COVID-19 ICU and expectably more for respiratory infection. Diagnosing hospital or ventilator-acquired pneumonia was challenging in which overdiagnosis and overtreatment were commonly practiced [39]. Respiratory sampling was less preferable in ventilated COVID-19 patients due to the concern of aerosolized transmission from the ventilator circuit, leading to a reduction in frequency and quality of microbiological investigation [40]. This further increased the uncertainty in infection diagnosis as well as lessened the reliance on microbiological results [41]. An international survey by Beovic et al. [42] reported that the preference for broad-spectrum antibiotics in COVID-19 patients and the decision on antibiotic prescription are mainly based on clinical presentation. However, it is challenging to differentiate bacterial etiology from COVID-19 pneumonia. Clinicians would proceed to escalate therapy when the patient’s progress was not satisfactory [43]. Moreover, broadening the antibiotic spectrum in managing infection of the critically ill could be a reaction to the fear of missing diagnosis, which needs to be addressed [39,44]. To date, there is no standard recommendation to guide escalation therapy. The current guidelines often recommend the initial choice but lack the guidance on next option when the patient worsens or is not progressing well. The usual practice is mainly broadening the spectrum of the antibiotic while pending microbiological reports [45]. Teitelbaum et al. [46] suggested employing antibiogram to guide the next empirical agents [46]. The study found that the escalation antibiogram did not support the usual exercise of switching from ceftriaxone to ceftazidime or piperacillin/tazobactam among ceftriaxone resistant GNR, but meropenem or amikacin instead. Predictably, this appears to encourage carbapenems prescription when ceftriaxone therapy fails. However, this approach should be applied on the caveat that antibiogram was derived from positive cultures and might not apply to all infections.

During the pandemic period, both the COVID-19 ICU and the GICU experienced a shortage in staffing as they were managed by the same clinician teams. Apart from the uncertainty in COVID-19 management, overwhelming workload and exhaustion could cause clinicians to rely on broad-spectrum antibiotics in the dread of missing possible infecting microbes [44]. However, this appeared to be true in only a small proportion of prescriptions evaluated as definitive therapy for ESBL organisms. When pathogens such as MDR *Acinetobacter* spp. and *Stenotrophomonas maltophilia* are isolated, carbapenems might be inadequate. Carbapenem-resistant *Acinetobacter baumannii* (CRAb) was isolated in substantial proportion among positive cultures from the COVID-19 ICU, compared to the GICU. This was consistent with studies by Rangel et al. [47] and Russo et al. [48], which noted that the incidence of CRAb was heightened among COVID-19 patients. According to the recent treatment guideline by IDSA, high dose ampicillin-sulbactam could be considered, but in reality, there is no antibiotic proven to be effective [49]. A cohort study among ICU patients found that mortality risk was further increased to twice as high for bloodstream infection without adequate therapy within the first 24 h [50]. Although carbapenems might have a role when used as a third agent in combination with ampicillin-sulbactam and polymyxin, this suggestion is based on in vitro studies and remains to be proven by robust clinical studies. The present results indicated that carbapenems were continued as empirical therapy when CRAb was isolated, for which a combination of high dose ampicillin-sulbactam at 9 g every 8 h with polymyxin is the recommended therapy by the current local ICU guideline [51]. Referring to the pathogens isolated from the blood cultures, carbapenems were overly broad for more than three-quarters of empirical prescriptions. Both inadequate and overbroad antibiotic spectrum could lead to poorer survival rates in patients at the odds of a 20% increase in mortality, as revealed in a large cohort study among US hospitals by Rhee et al. [52]. It was beyond the scope of the current study to correlate the association with mortality. However, this highlighted the need for enhancing antimicrobial stewardship and rapid diagnostic tools so that appropriate therapy could be optimized or deescalated promptly.

When ESBL-producing organism(s) is isolated, carbapenem is the preferred choice as there is yet an alternative agent proven non-inferiority as in the case of piperacillin/tazobactam in bacteremia [53]. While the empirical initiation of carbapenem might be rational considering the ESBL acquisition risk and unsatisfactory response requiring escalation, the duration was questionable. This study revealed that the carbapenems were empirically continued for about one week and the COVID-19 ICU had a longer course duration than the GICU. This was shorter than the median eight days in five French ICUs [32]. A recent position statement from European Societies of Intensive Care Medicine (ESICM), Clinical Microbiology and Infectious Diseases (ESCMID) advocates that daily review of antibiotics and de-escalation to narrower spectrum antibiotics should be performed for the critically ill according to microbiological results. Several studies supported that de-escalation is safe and associated with lower mortality [54]. When no growth is detected, the non-infectious cause should be investigated and antibiotics may be stopped [45,54]. The initiation and continuity of broad-spectrum carbapenems despite microbiological reports suggesting viable alternatives are concerning, as the risk of developing resistance increases endlessly by 2% for each day of meropenem exposure [7,55]. One of the possible explanations could be the time lapse required for the microbiological reports. In general, it took about two to four days to have organism identification and susceptibility reports from cultures [56]. Molecular methods such as multiplex polymerase chain reaction (PCR) and microarray or matrix-assisted laser desorption/ionization-time of flight mass spectrometry (MALDI-TOF MS) allow for rapid identification of organisms and resistance determination within hours, which would potentially enable clinicians to optimize antibiotic earlier [57]. Several studies showed that rapid testing, together with AMS, improves the time to appropriate antibiotics [58,59] and can potentially lead to better patient outcomes [60].

Negative cultures are also common among the critically ill. It usually takes five days of blood culture incubation before confirming negative growth [61], during which clinicians might choose to continue antibiotics before the report is finalized. A shorter incubation time might allow earlier decision-making on antibiotic prescription. An incubation period of up to four days [61] or even one day [62] might be possible with certain modern blood culture systems [61,62], which are often unavailable in resource-limited settings. Biomarkers including procalcitonin (PCT) could be used to guide the duration of therapy; however, a rise in PCT in the absence of microbiological evidence might compel the escalation or initiation of antibiotics [63,64] due to the knowledge gap and skepticism on PCT over clinical judgment [65,66], especially among COVID-19 patients who are critically ill and given concurrent steroid and tocilizumab [67].

Antibiotic prescription is often executed by focusing on the immediate benefit instead of the potential detrimental effect in the distant future, which was described by Langford et al. [68] as cognitive bias. Clinicians might prefer maintaining broad-spectrum antibiotics as a “safe option” despite the microbiological reports [43,69]. The perception and attitude could be a consequence of a deficiency in education and training during medical residency and undergraduate years [70]. Education is one of the objectives of the WHO global action plan for AMR [8]. Therefore, AMR and AMS modules should be part of training in critical care practitioners [71] who could act as synergistic AMS champions in ICU management. These would cultivate confident and judicious antibiotic prescribers [72] who are the key to combat against AMR, which is aggravated by antibiotics exposure [10].

The findings add to the existing paucity of information on exposure to broad-spectrum antibiotics in critical care areas in the South-East Asia region. We have demonstrated that the excessive antibiotic consumption is likely a result of unwarranted empirical use over a prolonged period and de-escalation is not performed promptly. Furthermore, we report the duration of therapy adjusted to indication, which provides more meaningful feedback to critical care clinicians for engagement in AMS initiatives. The same measurement can be adopted as a benchmarking across institutions and to design a standard tool of appropriateness assessment, which is currently lacking for critical care areas [71,73]. MDR organisms rate and prescription appropriateness in ICUs should be listed in the critical care units benchmarking worldwide [74] and be added as one of the foci of the global surveillance on antimicrobial resistance initiatives [75,76].

There were many limitations due to the nature of the retrospective observational study based on a single center. Furthermore, the data were retrieved from datasets focusing on carbapenems prescriptions and might not provide the whole picture of antibiotic prescription practice. There could also be missing data that were likely lost due to limited physical access to the COVID-19 ICU. The indication and prescription duration for COVID-19 ICU was extracted during table round discussion and, therefore, subjected to recall bias though data availability and accuracy became better when documents were made available electronically. This study reflected the practice during the COVID-19 pandemic year, which might be different from the usual practice before that. In addition, therapy was evaluated according to the microbiological reports and did not assess the correlation with infection severity [77]. However, this study appraised the reason for carbapenems prescription, which was closer to identifying that prescriber intention as a clinical judgment of severity could be subjective [39].

This current study provides a snapshot of the difference in the prescription practice and the microbiological profile among patients prescribed carbapenems between the COVID-19 ICU and the GICU. This is important as AMS strategies should cater to the circumstances under which broad-spectrum antibiotics are used [78]. ICU could be the epicenter for the spread of MDR organisms that are associated with higher patient mortality and the situation worsens with the pandemic. The AMS efforts should couple with infection control measures such as hand hygiene, resistance tracking, and transmission prevention to work synergistically in improving infection prevention and antibiotic use [79,80], to be better prepared for the ongoing and future pandemic wave(s). Further study should be done to identify risk factors and determine the consequence of carbapenem use on resistance trends and patient outcomes. The current data should alert the government and healthcare institutions to prioritize the effort in optimizing antibiotics use in ICUs. There is an urgent need to improve the epidemiological reporting and infrastructure for rapid microbiological diagnostics and reliable biomarkers, in addition to effective communication and knowledge dissemination to guide antibiotic prescription and exercise de-escalation early.

## 4. Materials and Methods

### 4.1. Study Design and Settings

This was a cross-sectional retrospective observational study conducted at the Hospital Canselor Tuanku Muhriz (HCTM), a 1054 bedded tertiary care university hospital located in Kuala Lumpur, Malaysia. This study included antibiotic prescriptions dispensed to ICU(s) during the period from 2016 to 2021. The unit used to be a 17-bedded medical/surgical ICU. In 2020, another ward was repurposed as COVID-19 ICU with only 3 beds initially. Following the worsening of the COVID-19 pandemic, the total ICUs’ capacity was configured as the COVID-19 ICU operated fully and expanded to be 22-bedded, while the GICU was 8-bedded since December 2020.

The GICU was a mixed medical and surgical-based intensive care unit and the COVID-19 ICU was designed specifically for patients with confirmed COVID-19 infection. The triaging for admission was based on the admission and discharge protocol of the local institution, which was adapted from Malaysia National Protocol [81] and criteria proposed by Malaysia Society of Intensive Care (MSIC) [82,83,84,85]. The severity of patients infected by the COVID-19 virus was categorized into 5 clinical stages from asymptomatic to severely ill based on syndromes [81]. Those who were admitted to COVID-19 ICU were stage 4 (symptomatic with pneumonia requiring supplemental oxygen) or stage 5 (critically ill with multi-organ derangement) or those with medical and surgical conditions that required ICU care with concomitant COVID-19 infection. The severity of illness was assessed using APACHE II score [86] upon admission to GICU only.

The GICU and COVID-19 ICU were primarily managed by clinicians of specialty in anesthesiology and intensive care. One ICU pharmacist was assigned to deliver pharmaceutical care service by participating in the daily handover rounds/discussions with a team remotely for COVID-19 ICU and performing bedside reviews for GICU. Medications were prescribed by medical officers on duty in both ICUs. 

The ICUs practiced a routine infection control measure of collecting nasal and rectal swabs from newly admitted patients. All microbiological investigations were done by an in-house microbiological diagnostic laboratory service. Organisms were identified by automated VITEK^®^ 2 system (bioMérieux, Marcy-l’Etoile, France). Antibiotic susceptibility testings (AST) were performed using the Kirby–Bauer disk diffusion method and results were interpreted according to Clinical and Laboratory Standard Institute (CLSI) guidelines [87].

Antibiotics were electronically prescribed using the hospital electronic prescription system Medipro^®^ to initiate the dispensing process based on the unit of use system by the pharmacy. Meanwhile, the administration of antibiotic(s) was manually documented using a paper-based prescription with columns for prescribers to note the indication of the antibiotic as empirical, definitive, or prophylaxis, and columns for administration by nurses for up to 7 days. Both electronic and manual prescriptions were renewed if the duration of antibiotic was beyond 7 days. Antibiotics prescriptions were guided by the national ICU antimicrobial prescribing guide [51] and local hospital ICU-specific antibiotic treatment protocol, which was based on a local antibiogram introduced in 2016. The dosage regimes in the GICU and the COVID-19 ICU were based on the same principle, including prolonged infusion and renal adjustment [51], as COVID-19 infection is not known to affect antibiotic pharmacokinetics [88]. Antibiotics including broad-spectrum beta-lactams such as piperacillin/tazobactam, cefepime, meropenem, and polymyxins were readily available as limited floor stock to administer the first dose. However, the subsequent continuation of carbapenems or piperacillin/tazobactam required specialists’ consent and authorization, whereas initiation of polypeptides required consent from the infectious disease consultant on duty. Hence, the consent was obtained using a paper-based pre-authorization form stating the indication and duration completed with signatures by relevant specialists to be submitted to the pharmacy department for screening and dispensing. Beginning from 2021, during daily work routine, for each carbapenem prescription, the ward pharmacist would document further details, which include the type of patients/infections, prescription indication (definitive/empirical/prophylaxis or from infectious disease consultation), the reason for empirical initiation, suspected site of infection, date of initiation and completion, and mortality during therapy. The dataset of the antibiotics, pre-authorized forms, and carbapenems monitoring details were kept in the pharmacy department. 

### 4.2. Data Collection

Data on antibiotics consumption were extracted from the manually recorded dispensing documents from the pharmacy department. The cumulative admissions and patient days data were acquired from the hospital department of health information. Prescriptions of carbapenems were extracted from antibiotics preauthorization forms and carbapenems monitoring datasets in the pharmacy department. Microbiological reports were accessed using the hospital’s online microbiological reports system (OMS). The duration of carbapenems therapy was calculated by subtracting the date of initiation from the date of completion and adjusted by adding one day. All carbapenems prescriptions for patients admitted to COVID-19 ICU and GICU wards during 2021 were included. Carbapenems prescriptions of patient(s) who died or were transferred out before pharmacist review were excluded due to incomplete data. Carbapenems courses during which the patient(s) died before doctors’ order to stop/complete therapy were excluded from the evaluation of therapy duration. 

### 4.3. Antibiotic Utilization

With reference to WHO methodology [14], the DDD used to estimate the parenteral antibiotic utilization was standardized according to the latest updated value. Therefore, the DDD for the commonly used antibiotics are amoxicillin/clavulanic acid: 3 g; ampicillin/sulbactam: 6 g; ceftriaxone: 2 g; cefepime: 4 g; piperacillin/tazobactam: 14 g; imipenem: 2 g; meropenem: 3 g; vancomycin: 2 g; polymyxin B: 0.15 g; colistin: 9 g. The utilization is estimated by the cumulative data based on the number of vials dispensed as follows:$$\mathrm{Number}\;\mathrm{of}\;\mathrm{DDD}\;\mathrm{for}\;\mathrm{the}\;\mathrm{year}\;=\frac{\mathrm{Total}\;\mathrm{number}\;\mathrm{of}\;\mathrm{dispensed}\;\mathrm{vials}\;\times \;\mathrm{strength}\;\mathrm{of}\;\mathrm{vial}\;\mathrm{in}\;a\;\mathrm{year}\;\left(g\right)}{\mathrm{DDD}\;\left(\mathrm{from}\;\mathrm{WHO}\right)}$$
$$\mathrm{Number}\;\mathrm{of}\;\mathrm{DDD}\;\mathrm{per}\;1000\;\mathrm{patient}\;\mathrm{days}=\frac{\mathrm{Total}\;\mathrm{number}\;\mathrm{of}\;\mathrm{DDD}\;\mathrm{for}\;\mathrm{the}\;\mathrm{year}}{\mathrm{Total}\;\mathrm{patient}\;\mathrm{days}\;\mathrm{for}\;\mathrm{the}\;\mathrm{year}}\times 1000$$

Antibiotic usage before the pandemic was estimated for the year 2016 to 2019. The antibiotic usage during the pandemic was estimated for 2021. The utilization during 2020 did not belong to either group due to the transitional operation of the COVID-19 ICU.

### 4.4. Definition

#### 4.4.1. Definitive/Empirical Prescribing

Carbapenems prescriptions were considered definitive when it was initiated or continued following the availability of microbiological results, with pathogen or susceptibility requiring coverage with carbapenems’ spectrum, from cultures of relevant sites except those from nasal swab and/or rectal swab for infection control surveillance purposes. If carbapenem-resistant *Enterobacterales* were isolated, carbapenem was considered indicated when MIC was less than 8 [89]. Conversely, empirical therapy was considered when carbapenems were initiated for presumed infection, continued, or completed without microbiological evidence [90] or the isolate(s) were susceptible to other beta-lactam antibiotics of a narrower spectrum, such as penicillins, second/third/fourth generation cephalosporins and/or penicillin/inhibitors; or the isolate(s) was resistant where carbapenems were deemed unsuitable. Empirical escalation was considered when carbapenems were switched from ongoing narrower spectrum beta-lactam therapy or added to ongoing antibiotic(s) therapy due to unsatisfactory response.

#### 4.4.2. Classification of Patient Types

Patient types were classified according to the risk factors of infection by resistant organisms. Type-1 or community-acquired infection referred to young patients with no or few comorbid conditions who had no contact with the health care system and no prior antibiotic treatment in the last 90 days; Type-2 or healthcare-associated infection referred to patients who had contact with the healthcare system in the past 3 months or less than 1 week in the hospital or less than 48 h in the ICU (e.g., admission into hospital or nursing home), had an invasive procedure or recent antibiotic therapy in the last 3 months or were more than 65 years old with few comorbidities [91,92]; Type-3 or nosocomial infections referred to patients who had hospitalization more than 5 to 7 days with or without infections following major invasive procedures or had recent and multiple antibiotic therapies or were more than 65 years old with multiple comorbidities (e.g., structural lung disease, immunodeficiency) [93].

#### 4.4.3. ESBL GNB Risk

The risk of infection with ESBL GNB was considered when a patient had received antibiotics in the past 90 days, especially second and third generation cephalosporins; hospitalization for more than 2 days in the past 90 days; was a resident in a nursing home; had chronic dialysis in the past 1 month; had home wound care, immunosuppressive disease, and/or therapy, catheter colonized by ESBL GNB and rectal swab with ESBL GNB. This was adapted from local guidelines [94,95].

### 4.5. Statistical Analysis

Antibiotic utilization was measured in units of DDD per 1000 patient days in aggregate annual data. All analyses were carried out using Statistical Package for the Social Sciences (SPSS), version 27.0 (IBM Corp, Armonk, NY, USA for descriptive analysis (percentage and frequency), categorical, and continuous data variables. Univariable analyses were performed with Chi-Squared test or Fisher Exact test to compare categorical variables where appropriate. The median of continuous variables was compared using the Mann–Whitney test. A *p*-value of <0.05 was used as the level of significance.

## 5. Conclusions

Antibiotics’ consumption in ICU increased markedly during the pandemic year, with near to two-fold increments in carbapenems utilization. Most carbapenem therapies were empirical and the reasons for prescribing differed between the two ICUs. Carbapenems were frequently prescribed to escalate therapy in the COVID-19 ICU, while in the GICU, it was for concern of ESBL GNB risk. Both ICUs had a similar duration of empirical carbapenems’ usage.

## Figures and Tables

**Figure 1 antibiotics-11-01172-f001:**
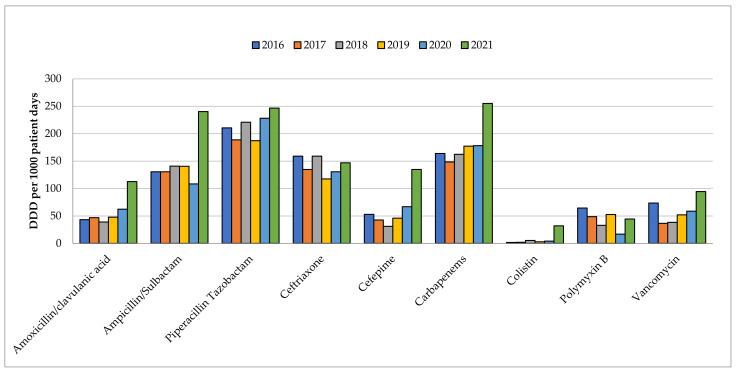
Annual Antibiotic Utilization in COVID-19 ICU and GICU from the year 2016 to the year 2021.

**Figure 2 antibiotics-11-01172-f002:**
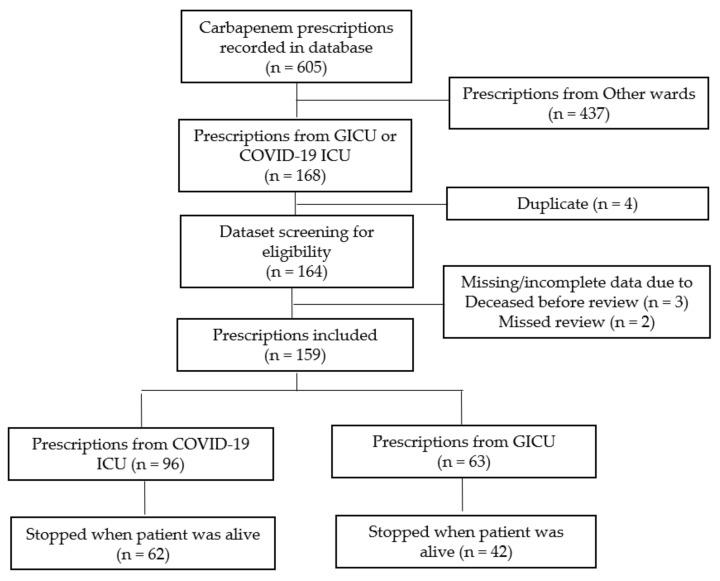
The selection process for eligible carbapenems prescriptions.

**Table 1 antibiotics-11-01172-t001:** Annual consumption of carbapenems in COVID-19 ICU and GICU.

	2016	2017	2018	2019	2020	2021	
**Annual Census**							
Number of Admissions, no	849	865	842	794	690	567	
Patient days, day	4636	5422	5504	5605	4228	6229	
Average length of stay, day	5.62	6.33	6.65	7.08	6.15	8.02	
**Carbapenems**	**Consumption (DDD per 1000 patient days)**						**Increment in 2021 versus 2019 usage (%)**
Ertapenem	12.66	−65.5	7.09	14.27	15.37	4.93	−65.5
Imipenem	43.67		15.99	27.25	12.89	23.16	
Meropenem	107.63		139.29	135.65	150.03	227.04	
Group-2 Carbapenems	151.30	53.6	155.28	162.90	162.92	250.20	53.6

**Table 2 antibiotics-11-01172-t002:** Rectal colonization among patients who were prescribed carbapenems.

Rectal Colonization by ESBL/MDR	Overall, n = 149	COVID-19 ICU, n = 91	GICU, n = 58	*p*
n (%)				0.003 ^a,^*
Yes ^	50 (35.7)	22 (25.9)	28 (50.9)	
No	90 (64.3)	63 (74.1)	27 (49.1)	
Unknown ^#^	9	6	3	

^#^ This group is not included in the analysis as rectal swab is not done; ^ 1 is *Citrobacter* spp.; ^a^ Pearson Chi-square. MDR, Multidrug-resistant. * *p* < 0.05 indicates statistically significant.

**Table 3 antibiotics-11-01172-t003:** Characteristics of all carbapenem prescriptions.

	Overall, n = 159	COVID-19 ICU, n = 96	GICU, n = 63	*p*
**Patient types at the time of prescription, no (%)**				0.033 ^a^
Type-1 (CA)	6 (3.8)	2 (2.1)	4 (6.3)	0.215 ^b,^^
Type-2 (HA)	26 (16.4)	11 (11.3)	15 (23.8)	0.039 ^a,^*^,^^
Type-3(NI)	127 (79.9)	83 (86.5)	44 (69.8)	0.011 ^a,^*^,^^
**Carbapenem, no (%)**				<0.001 ^b,^*
Meropenem	148 (93.1)	95 (98.9)	53 (84.1)	
Imipenem	8 (5.0)	0 (0.0)	8 (12.7)	
Ertapenem	3 (1.9)	1 (1.0)	2 (3.2)	
**Indication, no (%)**				0.310 ^a^
Definitive	21 (13.2)	11 (11.5)	10 (15.9)	
Empirical	138 (86.8)	85 (88.5)	53 (84.1)	

CA, Community-Acquired infection; HA, Healthcare-associated Infection; NI, Nosocomial Infection; ^a^ Pearson Chi-square, ^b^ Fisher Exact test. ^ based on individual groups. * *p* < 0.05 indicates statistically significant.

**Table 4 antibiotics-11-01172-t004:** Characteristics of empirical carbapenem prescriptions.

Reason for Empirical Therapy, no (%)	Overall, n = 138	COVID-19 ICU, n = 85	GICU, n = 53	*p*
Therapy escalation/switch	73 (52.9)	55 (64.7)	18 (34.0)	<0.001 ^a,^*
Considering ESBL GNB risk	43 (31.2)	19 (22.4)	24 (45.3)	0.005 ^a,^*
With ID consultation	10 (7.2)	7 (8.2)	3 (5.7)	0.741 ^b^
Others	12 (8.7)	4 (4.7)	8 (15.1)	0.059 ^b^
**Suspected site of infection, no (%)**				
Blood	45 (32.6)	31 (36.5)	14 (26.4)	0.220 ^a^
Central nervous system	6 (4.3)	3 (3.5)	3 (5.7)	0.675 ^b^
Intra-abdominal	20 (14.5)	4 (4.7)	16 (30.2)	<0.001 ^a,^*
Respiratory	62 (44.6)	45 (52.9)	17 (32.1)	0.017 ^a,^*
Skin and soft tissue	1 (0.7)	0 (0.0)	1 (1.9)	0.384 ^b^
Urinary tract	2 (1.4)	2 (2.3)	0 (0.0)	0.523 ^b^
Unknown	2 (1.4)	0 (0.0)	2 (3.8)	0.146 ^b^

^a^ Pearson Chi-square, ^b^ Fisher Exact test. * *p* < 0.05 indicates statistically significant.

**Table 5 antibiotics-11-01172-t005:** Microbiological profiles and organisms isolated prior to carbapenems therapy.

	Overall, n = 159	COVID-19 ICU, n = 96	GICU, n = 63	*p*
**Growth from cultures** ^1^				0.952 ^a^
Negative	10 (6.3)	6 (6.3)	4 (6.3)	
Mixed growth/*Candida* spp. ^2^	48 (30.2)	30 (31.3)	18 (28.5)	
Positive culture	101 (63.5)	60 (62.5)	41 (65.1)	
Site of positive cultures (n = 101)				0.736 ^a^
Positive blood cultures	66 (41.5)	40 (41.7)	26 (41.3)	
Other sites	35 (22.0)	20 (20.8)	15 (23.8)	
**Organisms isolated from blood cultures**				
**Definitive therapy:**				
*Escherichia coli* ESBL	4 (4.8)	1 (2.0)	3 (9.4)	
*Klebsiella pneumoniae*/spp. ESBL	9 (10.8)	5 (9.8)	4 (12.5)	
*Klebsiella pneumoniae ^#^* CRE	1 (1.2)	1 (2.0)	0 (0.0)	
*Pseudomonas aeruginosa* **	1 (1.2)	1 (2.0)	0 (0.0)	
*Achromobacter Xylosoxidans*	1 (1.2)	0 (0.0)	1 (3.1)	
**Empirical therapy:**				
*Escherichia coli*	4 (4.8)	0 (0.0)	4 (12.5)	
*Klebsiella pneumoniae*/spp.	7 (8.4)	2 (3.9)	5 (15.6)	
*Enterobacter aerogenes*/spp.	1 (1.2)	0 (0.0)	1 (3.1)	
*Acinetobacter baumannii*/spp.	1 (1.2)	0 (0.0)	1 (3.1)	
*Acinetobacter baumannii*/spp. MDR	9 (10.8)	8 (15.7)	1 (3.1)	
*Burkholderia cepacia*	1 (1.2)	1 (2.0)	0 (0.0)	
*Pseudomonas aeruginosa* ***	5 (6.0)	3 (5.9)	2 (6.3)	
*Stenotrophomonas maltophilia*	3 (3.6)	3 (5.9)	0 (0.0)	
*Enterococcus faecium/faecalis*/spp.	5 (6.0)	3 (5.9)	2 (6.3)	
*Streptococcus* spp.	3 (3.6)	1 (2.0)	2 (6.3)	
CoNS	13 (15.7)	10 (19.6)	3 (9.4)	
*Candida* spp.	5 (6.0)	4 (7.8)	1 (3.1)	
Others	10 (12.0)	8 (15.7)	2 (6.3)	
**Organisms isolated from respiratory/** **tissue/pus/urine cultures**				
**Definitive therapy:**				
*Escherichia coli* ESBL	1 (2.0)	1 (3.6)	0 (0.0)	
*Klebsiella pneumoniae*/spp. ESBL	3 (6.0)	2 (7.1)	1 (4.5)	
*Klebsiella pneumoniae ^#^* CRE	1 (2.0)	0 (0.0)	1 (4.5)	
*Enterococcus* spp.	1 (2.0)	1 (3.6)	0 (0.0)	
CoNS	1 (2.0)	1 (3.6)	0 (0.0)	
**Empirical therapy:**				
*Escherichia coli*	2 (4.0)	1 (3.6)	1 (4.5)	
*Klebsiella pneumoniae*/spp.	5 (10.0)	1 (3.6)	4 (18.2)	
*Klebsiella pneumoniae*^##^ CRE	1 (2.0)	1 (3.6)	0 (0.0)	
*Enterobacter aerogenes*/spp.	2 (4.0)	1 (3.6)	1 (4.5)	
*Acinetobacter baumannii*/spp. MDR	14 (28)	10 (35.7)	4 (18.2)	
*Pseudomonas aeruginosa* ***	10 (20.0)	5 (17.9)	5 (22.7)	
*Stenotrophomonas maltophilia*	2 (4.0)	1 (3.6)	1 (4.5)	
*Enterococcus faecium/faecalis*/spp.	2 (4.0)	2 (7.1)	0 (0.0)	
*Staphylococcus aureus*	2 (4.0)	0 (0.0)	2 (9.1)	
MRSA	1 (2.0)	0 (0.0)	1 (4.5)	
*Mycobacterium tuberculosis*	2 (4.0)	1 (3.6)	1 (4.5)	

^a^ Pearson Chi-square; CoNS: coagulase-negative *Staphylococci*; CRE, carbapenem-resistant *Enterobacterales*; ESBL, extended-spectrum beta-lactamase; MIC, minimum inhibitory concentration; MDR, multidrug-resistant; MRSA, methicillin-resistant *Staphylococcus aureus*. ^#^ MIC = 4; ^##^ MIC more than 24 for all carbapenems tested; ** Resistant to ceftazidime, cefepime, and piperacillin/tazobactam; *** Sensitive to ceftazidime. ^1^ Based on cultures reports prior to carbapenems therapy. ^2^ From tracheal aspirates/sputum/urine/pus.

**Table 6 antibiotics-11-01172-t006:** Duration of carbapenems prescriptions.

**Duration of Therapy**	**Overall, n = 104**	**COVID-19 ICU, n = 62**	**GICU, n = 42**	** *p* **
Overall median, days (IQR)	7 (5–8)	7 (5–8)	7 (4–9)	0.963 ^a^
Definitive therapy, median, days (IQR)	(n = 15) 8 (7–11) *	(n = 8) 8 (7–8)	(n = 7) 9 (7–14)	0.463 ^a^
Empirical therapy, median, days (IQR)	(n = 89) 7 (4–8)	(n = 54) 7 (5–8)	(n = 35) 6 (4–8)	0.654 ^a^

^a^ Mann Whitney; * *p* = 0.010; IQR, interquartile range.

## Data Availability

All data generated and analyzed during this study are included in this article.

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
