# Peer review of "Plethora of Antibiotics Usage and Evaluation of Carbapenem Prescribing Pattern in Intensive Care Units: A Single-Center Experience of Malaysian Academic Hospital"

_antibiotics, 2022, doi:10.3390/antibiotics11091172_

Round 1
Reviewer 1 Report
Many thanks for the opportunity to review this interesting and well presented paper. Authors touch an important issue: the over prescription of antibiotics with high risk in antimicrobial resistance.
Congratulations for paper. Below my minor suggestions that hope can contribute a little to improve your already good paper.
Introduction: introduce better the antimicrobial resistance as global health problem wordwilde that increase also during covid 19 pandemic (see and cite Impact of SARS-CoV-2 Epidemic on Antimicrobial Resistance: A Literature Review. Viruses. 2021 Oct 20;13(11):2110. doi: 10.3390/v13112110)
Methods and results: clear and well presented
Discussion: discuss better on the role of education on this items also during medical degree (see and cite Italian young doctors' knowledge, attitudes and practices on antibiotic use and resistance: A national cross-sectional survey. J Glob Antimicrob Resist. 2020 Dec;23:167-173). Education on this item can contribute in a strong way to control AMR spread
Furthermore, add also the risk to "burn" the new molecola for gram negative and gram positive if we continue with spread of AMR and not have a good education on the correct use
Finally, underline the role of IPC (infection prevention control) to contain use of antibiotic and reduce hospital infection
Moreover, add some global health proposal that came from your interesting paper
Author Response
Thank you for the response. We hope that our changes, as well as in this reply have helped to clarify the issues pointed out by the reviewers. We have answered the issues using published references and scientific claims.
Attached to this is a revised draft of our manuscript indicating the changes. All authors have agreed to the contents of the manuscript in its revised form.
Please see the attached

Reviewer 2 Report
This is a good observational retrospective study, and this manuscript can be considered for publication based on the answering reasonably the following queries.
Line 79
Did you define the COVID-19 ICU and GICU and physiologically score based critical illness? Level of seriousness of COVID19?
Line 89
Did you discussed the reason of polymyxin B dropped?
Line 103
In this table 1 patient admitted in 2020 and 2021 were less than other previous years.
General
Do your data on rising resistance of carbapenems in Malaysia?
Discuss more on, how escalation and de-escalation performed by the physicians.
What was reason aminoglycosides were not prescribed in your current study?
What is the advantage of this study and how rational is heavily usage of carbapenems in ICU?
Do you have data on Antimicrobial Sensitivity Test (AST) and respective MIC for different antibiotics?
How (criteria) were antibiotic doses decided for COVID-19 patients?
Author Response
Thank you for the review. We hope that our changes, as well as in this reply have helped to clarify the issues pointed out by the reviewers. We have answered the issues using published references and scientific claims.
Please see the attachment for the responses and the revised draft of our manuscript indicating the changes in red. The paper has been carefully revised to improve the grammar and readability.
